# *Suaeda salsa* NRT1.1 Is Involved in the Regulation of Tolerance to Salt Stress in Transgenic *Arabidopsis*

**DOI:** 10.3390/ijms241612761

**Published:** 2023-08-14

**Authors:** Yi Xiong, Saisai Wang, Cuijie Cui, Xiaoyan Wu, Jianbo Zhu

**Affiliations:** College of Life Sciences, Shihezi University, Shihezi 832000, China; xyshzu2017@163.com (Y.X.); 18299091648@163.com (S.W.); ccj19987654321@163.com (C.C.); wxy1245891567@163.com (X.W.)

**Keywords:** *SsNRT1.1C*, *Suaeda salsa* (L.) Pall., SsHINT1, salt tolerance, K^+^/Na^+^

## Abstract

Like other abiotic stresses, salt stress has become a major factor that restricts the growth, distribution and yield of crops. Research has shown that increasing the nitrogen content in soil can improve the salt tolerance of plants and nitrate transporter (NRT) is the primary nitrogen transporter in plants. *Suaeda salsa* (L.) Pall is a strong halophyte that can grow normally at a salt concentration of 200 mM. The salt stress transcriptome database of *S. salsa* was found to contain four putative genes that were homologous to NRT, including *SsNRT1.1A*, *SsNRT1.1B*, *SsNRT1.1C* and *SsNRT1.1D*. The cDNA of *SsNRT1.1s* was predicted to contain open reading frames of 1791, 1782, 1755 and 1746 bp, respectively. Sequence alignment and structural analysis showed that the SsNRT1.1 amino acids were inducible by salt and have conserved MFS and PTR2 domains. Subcellular localization showed they are on the endoplasmic reticulum. Overexpression of *SsNRT1.1* genes in transgenic *Arabidopsis* improves its salt tolerance and *SsNRT1.1C* was more effective than others. We constructed a salt-stressed yeast cDNA library and used yeast two-hybrid and BiFC technology to find out that SsHINT1 and SsNRT1.1C have a protein interaction relationship. Overexpression of *SsHINT1* in transgenic *Arabidopsis* also improves salt tolerance and the expressions of Na^+^ and K^+^ were increased and reduced, respectively. But the K^+^/Liratio was up-regulated 11.1-fold compared with the wild type. Thus, these results provide evidence that SsNRT1.1C through protein interactions with SsHINT1 increases the K^+^/Na^+^ ratio to improve salt tolerance and this signaling may be controlled by the salt overly sensitive (SOS) pathway.

## 1. Introduction

As an abiotic stress, salt stress affects the normal growth and development of plants, sometimes resulting in death [1,2]. No stress more extensively restricts plant growth and development than that of salt. It is estimated that by the middle of the 21st century, approximately 50% of all cultivated land may be damaged by soil salinization [3]. Phenotypically, salt stress can inhibit the growth, development and yield of plants to varying degrees [4]. This can include blocking the absorption of water by the roots, inducing chlorosis causing the leaves to become smaller or wilt and even causing the plant to die in severe cases [5]. Regarding internal physiology, salt stress primarily causes osmotic stress and ion toxicity, as well as subsequent oxidative stress which eventually leads to the blockage of protein synthesis and decreases in the rates of photosynthesis and growth [6,7].

The effects of salt stress on plants can be divided into osmotic stress, oxidative stress, photosynthetic damage and ion toxicity [8,9,10]. The main inorganic ion in saline soil is primarily sodium (Na^+^). Owing to the similarity between K^+^ and Na^+^, many K^+^ transporters cannot distinguish between the two ions, which results in a high content of exogenous Na^+^ affecting the intake of K^+^ and in a higher content of Na^+^ than K^+^ in the cell [11]. Potassium (K) is one of the major elements necessary for plant growth and it is an important cation in cells [12]. It plays a vital role in maintain cell turgor, cell membrane potential and protein function [13]. In addition, the biosynthesis of more than 50 enzymes, including starch synthase, glutamine synthetase and pyruvate kinase, requires the activation of K^+^ [14,15]. Na^+^ will compete for the K^+^ binding site of related enzymes and reduces their activity which will further affect the biosynthesis of proteins, photosynthesis and other plant biochemical activities [16].

To adapt to this unfavorable environment, plants that have grown in this type of environment for a long time have evolved a series of unique mechanisms to resist the salt stress. Among them, the salt overly sensitive (SOS) and abscisic acid (ABA) signaling pathways have been the most comprehensively studied [17,18]. The SOS signaling pathway can enhance the salt tolerance of plants by increasing the ratio of K^+^/Na^+^ in cells [13]. Through the activation of the SOS signaling pathway, the excessive Na^+^ in cells is discharged to the extracellular environment and it maintains the balance of K^+^/Na^+^ in cells and regulates the adaptability of plants to salt stress [19]. Under salt stress, SOS2 needs to reach the cell membrane after being activated by SOS3 and then interacts with the membrane-localized Na^+^/H^+^ antiporter SOS1 to ensure the efflux of Na^+^ [20].

Nitrogen (N) is one of the major elements which is necessary for plant growth and development. N is not only the primary structural material of plants but also the most important component of various enzymes in cells, which play an important role in crop growth and yield [21]. Under salt stress, the application of a particular amount of N fertilizer can improve the salt resistance of plants [22]. The addition of a medium concentration of ammonium nitrate (NH_4_NO_3_) can promote the growth of *Suaeda salsa* under salt stress [23].

Plants primarily absorb organic and inorganic N through their roots. The N absorbed by most plants is primarily inorganic N and is typically either nitrate (NO_3_^−^) or ammonium (NH_4_^+^) [24]. Because most plants are primarily terrestrial, nitrification usually occurs in soil that contains sufficient oxygen, so that nitrate is the primary source of N for plant growth and development [25]. Nitrate transporter 1.1 (NRT1.1) is a member of the nitrate transporter 1 (NRT1) family which is the largest nitrate transporter family in plants [26].

*Suaeda salsa* (L.) Pall. is an annual herb plant of the Chenopodiaceae [27]. It is distributed in saline alkali soil and beaches. It is also a typical euhalophyte that is strongly salt tolerant and grows normally at concentrations of 200 mM NaCl [28]. Currently, two salt stress transcriptomes of *S. salsa* (PRJNA527358 and PRJNA512222) have been published [29,30]. We reorganized and mined the transcriptome data of *S. salsa*, enabling the coding sequence (CDS) region of *Arabidopsis AtNRT1.1* to be used for a local BLAST and four *SsNRT1.1* genes were finally obtained. Overexpression *SsNRT1.1* genes in *Arabidopsis* was used to functionally characterize their roles in salt tolerance. This study shows that the expression of *SsNRT1.1C* significantly enhances salt tolerance in *Arabidopsis* and primarily positively regulates salt stress by regulating the SsHINT1 protein.

## 2. Results

### 2.1. Structural Analysis of the NRT Proteins from Suaeda salsa and Arabidopsis

The CDS of *Arabidopsis AtNRT1.1* was used as a probe after organizing and analyzing the published transcriptome data. Four *SsNRT1.1* genes were obtained from *S. salsa*. We found SsNRT1.1 and AtNRT1.1 proteins were not very similar and homologies were 63.0%, 62.7%, 50.8% and 47.8%, respectively. These genes were designated *SsNRT1.1A*, *SsNRT1.1B*, *SsNRT1.1C* and *SsNRT1.1D* based on their level of homology. The relevant biological information of the *SsNRT1.1* genes is shown in Table 1.

Different homologous *NRT* genes were identified from plant species based on the NCBI protein database. Protein sequences of SsNRT1.1s from *S. salsa* and from other NRT proteins were analyzed for their phylogenetic relationship using ClustalW multiple sequence alignment. The four SsNRT1.1 proteins are located in different branches and have a distant evolutionary relationship with the model plants rice (*Oryza sativa* L.) and *Arabidopsis*. Among these four SsNRT1.1 proteins, SsNRT1.1A and SsNRT1.1B are relatively closely related evolutionarily, while SsNRT1.1C and SsNRT1.1D are also relatively closely related evolutionarily (Figure 1A). A structural analysis of the different SsNRT1.1 proteins showed that they harbored similar functional domains, which are the major facilitator superfamily (MFS) and peptide transport 2 superfamily (PTR2) (Figure 1B).

### 2.2. Differential Expression of SsNRT1.1 Genes in Response to Salt Stress in Suaeda salsa

To further clarify the expression of *SsNRT1.1* genes in different tissues, we extracted total RNA from the roots, stems and leaves of six-week-old *S. salsa* and reverse transcribed it into cDNA. A specific primer set was designed for real-time quantitative reverse transcription PCR (qRT-PCR). *SsNRT1.1A* was primarily expressed in the roots, and the levels of expression in the roots, stems and leaves did not change significantly. *SsNRT1.1B* is primarily expressed in the stems of *S. salsa*, while *SsNRT1.1C* and *SsNRT1.1D* are primarily expressed in its leaves (Figure 2A). 

To explore the involvement of the *SsNRT1.1* genes in response to salt stress, six-week-old leaves of *S. salsa* were cultivated and used as the experimental model. A concentration of 400 mM NaCl was chosen as the salt stress treatment. The RNA was extracted from the leaves at the time points 0 h, 3 h, 6 h, 9 h, 12 h and 24 h. The levels of expression of the four *SsNRT1.1* genes in *S. salsa* gradually increased over time following treatment with 400 mM NaCl (Figure 2B). Among them, the *SsNRT1.1C* gene showed the largest change and the level of expression of *SsNRT1.1C* increased the most by 3.05-fold at 12 h. However, after 24 h of salt stress, the levels of expression of all four *SsNRT1.1* genes began to decrease.

Nevertheless, salt stress results in differential gene expression in *SsNRT1.1*, which suggests that these genes may have a function in salt tolerance.

### 2.3. Subcellular Localization of the SsNRT1.1-GFP Fusion Protein

We constructed the C-terminal green fluorescent protein (GFP) fusion vector SsNRT1.1s-GFP to reveal the location of these genes in cells. GFP was used as the control and observed separately. The proteins of these four genes were found to be primarily distributed in the endoplasmic reticulum (ER) (Figure 3). This is different from previous research that found that AtNRT1.1 is primarily localized to the cell membrane, while OsNRT1.1A and OsNRT1.1B are primarily localized to the vacuolar membrane and cell membrane [31,32]. The same genes of different species are located in different organelles which strongly suggests that their functions are also different. The primary functions of the ER include protein synthesis, processing, energy metabolism, calcium ion storage and the regulation of redox signals in eukaryotic cells. Therefore, we hypothesize that the *SsNRT1.1* genes of *S. salsa* may be involved in protein synthesis, processing and energy metabolism in cells.

### 2.4. Overexpression of the SsNRT1.1 Genes in Transgenic Arabidopsis under Salt Stress

To explore the function of *SsNRT1.1s* in *S. salsa*, we constructed a plant overexpression vector and overexpressed it in *Arabidopsis*. Under normal conditions, transgenic *Arabidopsis* has a larger phenotype than that of the wild type (WT). This result shows that the *SsNRT1.1* can improve plant growth and increase its biomass (Figure 4). Thirty-day-old *Arabidopsis* plants were treated to 250 mM NaCl to induce salt stress and irrigated every 2 d to ensure that the weight of plants remained consistent. They were photographed after 5, 10 and 15 days. The contents of malondialdehyde (MDA) and proline (Pro), relative electronic rate (REC) and superoxide dismutase (SOD) were measured separately (Figure 4).

No significant changes in the WT and transgenic *Arabidopsis* were observed at 5 d. However, some leaves of the WT began to turn white at 10 d, while there was no significant change in the transgenic *Arabidopsis*. At 15 d, the WT had mostly turned white and could not grow normally, but the transgenic plants grew normally and had varying degrees of whitening on their leaves. In these transgenic lines, 35S:: SsNRT1.1C *Arabidopsis* had the most salt-tolerant phenotype.

Physiological indexes of different transgenic lines were tested separately. Under normal circumstances, there were no significant changes in the physiological indicators of *Arabidopsis*. However, as the duration of salt stress increased, different physiological indicators of each transgenic line began to change. In the overexpression lines, the changes in Pro were consistently greater than those in the WT plants and that of the *SsNRT1.1C* line was the highest. The largest increase in MDA and REC was observed in WT and transgenic *SsNRT1.1C* lines showed the lowest level of increase in these parameters. *Arabidopsis* had a higher SOD content at 5 d than on day 0 and transgenic plants had a higher SOD content than WT. On the 10th day, the SOD content of all *Arabidopsis* plants began to decrease, and that of WT decreased especially significantly. On the 15th day, the SOD content of WT was only 46% of the initial level and the transgenic lines’ contents were still above 73% (Figure 5). That of the *SsNRT1.1C* transgenic *Arabidopsis* was consistently higher than those of other transgenic lines. These results led to the conclusion that *SsNRT1.1C* is the most effective at improving salt tolerance.

### 2.5. The Expression of SsNRT1.1C in S. salsa under Different Levels of Salt Stress and Construction of a Yeast Membrane Library

In order to clarify the specific function of *SsNRT1.1C* in salt tolerance, we needed to study the functional proteins that interact with SsNRT1.1C. Because SsNRT1.1C has a transmembrane domain, we needed to construct a yeast membrane library of *S. salsa*.

Thirty-day-old seedlings of *S. salsa* were treated with 300, 400 and 500 mM NaCl to induce salt stress. The roots, stems and leaves of *S. salsa* were selected and treated for 0 h, 3 h, 6 h, 9 h, 12 h and 24 h.

The total RNA was extracted from the plants and then reverse transcribed for the qRT-PCR experiment. The levels of expression of *SsNRT1.1C* in different plant tissues varied with different concentrations of salt and time periods, but the overall changes in expression in leaves were always higher than in roots and stems. The level of expression of *SsNRT1.1C* in the leaves reached its maximum value at 12 h after treatment with 300 mM NaCl, which was 3.11-fold higher than in the control group. The levels of expression of *SsNRT1.1C* in the roots and stems reached their maximum values at 12 h and 9 h, which were 1.52- and 1.44-fold higher than in the control group, respectively. Under 400 mM NaCl treatment, the level of expression in the leaves reached its maximum at 12 h. This was 3.05-fold higher than in the control group and the level of expression in the roots and stems reached its maximum at 12 and 9 h, which was 1.71- and 1.73-fold higher than in the control group. Following treatment with 500 mM NaCl, the level of expression in the leaves reached its peak at 9 h, which was 3.34-fold higher than in the control group. The level of expression in the roots reached its peak at 6 h, which was 1.71-fold higher than in the control group. In contrast, the level of expression in the stems reached its peak at 9 h, which was 1.98-fold higher than in the control group.

We selected different concentrations and tissues with the highest levels of expression to construct a salt-stressed yeast cDNA library of *S. salsa* and identified the quality of the library. The final confirmed storage capacity of the yeast library was 8.76 × 10⁷ CFU·mL^−1^ with a total clone count of 1.752 × 10⁸ CFU. The rate of recombination was 95.8% (Figure 6).

### 2.6. Constructed SsNRT1.1C Bait Vector

We verified the self-activating ability of the constructed pBT3-N-SsNRT1.1C vector. No self-activation phenomenon was found on the SD/-Leu/-His block. Moreover, no expression of the *HIS3* gene was observed (Figure 7).

### 2.7. Yeast Two-Hybrid Screening and BiFC Validation

The NMY51 yeast strain that contained the bait plasmid pBT3-N-SsNRT1.1C was used as the receptor strain to prepare the competent. The yeast cDNA library plasmid was transferred into it and coated on a screening plate of SD-TLH with 0 mM of 3-amino-1,2,4-triazole (3AT). The positive monoclonal clones that were grown on the SD-TLH screening library plate were then crossed onto a 0 mM 3AT SD-TLH selection plate and incubated at 30 °C for 3–4 days (Figure 8).

Positive monoclonal clones that had been grown on SD-TLH-deficient plates were diluted with sterile water at concentrations of 10^0^, 10^−1^ and 10^−2^ and then incubated on plates deficient in SD-TL, SD-TLH and SD-TLHA at 30 °C for 3–4 days. As previously described, the positive control was selected as pBT3-STE-OsPT2 + pPR3-N-OsPP95, and the negative control was pBT3-STE-OsPT2 + pPR3-N-OsPP95CT [33]. The rotation validation results showed that all 12 monoclonal transformants grew monoclonal clones in the rotation experiment (Figure 9). Therefore, this experiment provides preliminary evidence that these 12 proteins may have an interactive relationship with SsNRT1.1C.

Owing to the possibility of false positives, we constructed bimolecular fluorescence complementation (BiFC) vectors for these 12 genes and conducted BiFC experiments with SsNRT1.1C, ultimately confirming the protein interaction relationship between SsHINT1 and SsNRT1.1C (Figure 10).

### 2.8. Overexpression of SsHINT1 in Transgenic Arabidopsis under Salt Stress

To explore the function of *SsHINT1*, we built a plant overexpression vector and overexpressed in it transgenic *Arabidopsis.*

After 30 days of normal growth, the transgenic plants were larger than the WT. This demonstrated that *SsHINT1*, like *SsNRT1.1C*, can improve plant growth and increase its biomass.

There were no significant changes in the phenotypes of WT and *SsHINT1* transgenic *Arabidopsis* after 5 d under 250 mM NaCl salt stress. However, one-third of the leaves of WT began to turn white after 10 d of salt treatment and transgenic plants only had a few leaf edges turn white. This indicates that salt stress damaged the WT but had not yet impacted the transgenic plants. More than half of the WT could not continue growing after 15 d and most of the surviving *Arabidopsis* leaves turned white. However, transgenic *Arabidopsis* had only a few white leaves (Figure 11).

These results showed that the WT had stopped growing by 15 d. The transgenic plants only began to show whitening and the plants could still grow.

Under normal circumstances, there were no significant changes in the physiological indicators of *Arabidopsis*. During salt stress treatment, the Pro content was consistently higher than in WT plants. The content of MDA and REC increased with salt stress in both WT and transgenic plants but the WT had higher levels than transgenic plants. On the 15th day, the transgenic plants’ contents of MDA and REC were only 61.2% and 69.5% those of WT, respectively. The content of SOD first increased and then decreased during salt treatment. The WT showed a smaller increase and a larger decrease compared to the transgenic plants. In particular, on the 15th day of salt stress, the SOD content in the WT was only 63.6% that of transgenic plants (Figure 12).

### 2.9. Expression and Ion Concentration Detection of Salt Resistance Genes Related to Salt Stress in Transgenic SsHINT1 Arabidopsis

RNA was extracted from *Arabidopsis* leaves that had been stressed by salt for 0, 5, 10 and 15 d. Specific primers based on the sequences of *AtSOS1*, *AtSOS2*, *AtSOS3*, *AtRD22* and *AtRD29A* were designed for use with qRT-PCR. During salt stress, the levels of expression of these five genes in the transgenic strains were higher than in WT. Except for the expression of *AtSOS1* which differed the most substantially and was 2.7-fold higher than in the control, the others had no significant difference (Figure 13).

The contents of Na^+^ and K^+^ in *Arabidopsis* were then measured separately. As the time of salt treatment increased, the concentrations of Na^+^ and K^+^ began to increase and decrease, respectively. The WT changed more dramatically than the transgenic lines. The K^+^/Na^+^ ratio began to decrease but the ratio in the transgenic line was consistently higher than that in the WT (Figure 14).

## 3. Discussion

The qRT-PCR results of the *SsHINT1* transgenic *Arabidopsis* salt treatment showed that the expression of salt resistance genes increased compared with the WT. In particular, *AtSOS1* was increased most highly by 2.7-fold compared to the WT. As AtSOS1 is an important functional protein in the SOS salt tolerance signaling pathway and can regulate the K^+^/Na^+^ ratio, we measured the contents of Na^+^ and K^+^ in Arabidopsis. We found that the ratio of K^+^/Na^+^ in the plants decreased during the salt treatment but the ratio in transgenic plants was always higher than in the WT. It was 11.1-fold higher at 15 d.

NRT1.1 is a transmembrane transport protein and mainly transports nitrogen-containing compounds such as NO_3_^−^, oligopeptides and amino acids. NRT1.1 contains the MFS and PTR2 domains. The primary functions of MFS and PTR2 are to promote the membrane transportation of solutes, such as sugars, drug molecules, peptides and citric acid cycle metabolites under an electrochemical gradient.

Previous research primarily focused on how NRT1.1 improves the efficiency of utilization of N and enhances the resistance of plants to metal ion stress, such as lead (Pb^2+^) and cadmium (Cd^2+^) stress [34,35]. The relationship between NRT1.1 and salt stress has not yet been reported. However, many different studies have shown that increasing the amounts of N in the soil can enhance salt tolerance. As NRT1.1 is the primary nitrate transport protein that functions in soil, it may enhance the tolerance of plants to salt.

Four NRT1.1 genes of *S. salsa* were obtained by rearranging and analyzing the transcriptome. It has more *NRT1.1* genes than model plants. These four genes are mainly expressed in roots, stems and leaves. The results of subcellular localization experiments showed that the four *SsNRT1.1* genes are all located in the ER. In contrast, *AtNRT1.1* is located in the cell membrane and *OsNRT1.1A* and *OsNRT1.1B* are primarily located in the vacuolar membrane and cell membrane. These results showed a certain difference in NRT1.1 between halophytes and non-halophytes and some extra functional differences between the NRT1.1 genes in *S. salsa* and those in non-halophytes.

We constructed an *SsNRT1.1* gene plant overexpression vector and transformed it into *Arabidopsis* to obtain the T_3_ generation. Under normal circumstances, the transgenic plants were larger than the WT, indicating *SsNRT1.1s* can enhance plant biomass, which is similar in function to the *NRT1.1* genes of other species. Salt stress treatment showed that the transgenic *Arabidopsis* is more tolerant than the WT and transgenic *SsNRT1.1C Arabidopsis* was the most effective at tolerating salt stress.

To further validate the experimental results, relevant physiological indicators were also tested. Plants maintain an osmotic balance under adverse conditions by accumulating osmotic regulatory substances. The accumulation of Pro, which is the most important osmotic regulatory substance in plants, can stabilize cellular structures and reduce oxidative losses in plants. The changes in MDA and REC can reflect the degree of damage of the plant cell membrane under stress conditions. SOD scavenges ROS, such as anions derived from O_2_. It can reduce the peroxidative damage to the cell membrane caused by salt stress. We further validated our results by measuring the four physiological indicators described above.

The results indicate that *SsNRT1.1* can enhance the ability of plants to tolerate salt. However, they still cannot explain the specific pathway of *SsNRT1.1C* to improve plant salt tolerance.

Thus, we analyzed the expression of *S. salsa* under different levels of salt stress. We selected samples from different tissues with the highest levels of expression and constructed a yeast cDNA library and SsNRT1.1C bait vector. The protein interaction between SsHINT1 and SsNRT1.1C was ultimately determined through Y2H screening and rotational and BiFC validation.

Research on *HINT1* has primarily been focused on animal models. Therefore, we constructed a plant overexpression vector and transformed it into *Arabidopsis* undergoing salt stress treatment. Finally, we determined that transgenic *Arabidopsis* had enhanced salt tolerance. Through qRT-PCR and ion concentrations, we found that the *SsHINT1* transgenic *Arabidopsis* had enhanced salt tolerance by increasing the K^+^/Na^+^ ratio and mainly achieved this by activating SOS signals.

Therefore, we hypothesized that SsNRT1.1C enhances the plant salt tolerance pathway. Under normal conditions, SsNRT1.1C and SsHINT1 proteins interact in the ER. When the plant cells are stimulated by salt stress, SsHINT1 leaves the ER and activates the SOS1/2/3 signaling pathway by entering the nucleus. After the SOS signaling pathway is activated, it enhances plant salt tolerance by increasing the K^+^/Na^+^ ratio (Figure 15).

## 4. Materials and Methods

### 4.1. Primers Used in the Experiment

All the primers used in the experiment are in Table 2.

### 4.2. Salt Stress Treatment of Suaeda salsa Seedlings in Soil

Five-week-old *S. salsa* plants that were 15.0 cm high were subjected to treatment with salt using 60 mL of 200, 300, 400 and 500 mM NaCl. The roots, stems and leaves of *S. salsa* were collected at 0, 3, 6, 9, 12 and 24 h after treatment, rapidly frozen with liquid nitrogen for 5 min and stored at −80 °C.

### 4.3. Search for SsNRT1.1 and Screening in Suaeda salsa

The published salt stress transcriptome data of *S. salsa* were downloaded from the NCBI. The raw data were filtered using FASTP v. 0.20.0 to remove sequencing adapters and low-quality and overly short reads to obtain clean data. The total bases and total reads were counted, Q30, Q20, GC content and effective data ratio were measured and a filter and FastQC were used to evaluate the quality of the data.

After the original data had been screened, Trinity v. 2.9.1 was used to assemble the transcript sequence and the longest transcript of each gene was used as the unigene. Statistics were conducted on the assembly results. Geneious Prime was used to probe the coding sequence (CDS) of *AtNRT1.1* (AT1G12110), which was used as a query sequence for a local BLAST analysis. Single-stranded complementary DNA (sscDNA) was selected for the database, TBLASTN for the program, 1e^−10^ for the maximum E-value and default values were used for the other options. The sequences obtained were sorted and duplicate sequences were removed. The candidate genes were compared and analyzed using DNAMAN.

### 4.4. qRT-PCR Experiment

The qRT-PCR reaction system in this experiment was 20 μL. Before the experiment, the cDNA concentration of each material was detected using NanoDrop spectrophotometry (Thermo Fisher, Waltham, MA, USA) and a standard of 600 ng of cDNA was uniformly added. The remaining volume was supplemented with ddH_2_O. *SsACTIN* was selected as the internal reference gene. The qRT-PCR reaction system is in Table 3.

### 4.5. Subcellular Localization

Arabidopsis seedlings were cultivated in soil for 3–4 weeks. Twenty leaves were collected and placed in a 50 mL triangular flask. They were then soaked in 5–10 mL of enzymatic hydrolysate, vacuumed and wrapped in sealing film. The mixture was incubated in low light and at 23 °C for 3 h, so that the enzymes could dissolve the cell walls. It was then filtered through 40 µm nylon gauze into a 2 mL Eppendorf tube. The filtrate was retained and centrifuged at 400 rpm for 5 min at 4 °C. The supernatant was removed and the sample was then washed twice with 1 mL of precooled W5 solution. The sample was centrifuged at 400 rpm for 5 min at 4 °C and then gently suspended in 1 mL of W5 solution. The samples were incubated on ice for 30 min and then centrifuged at 400 rpm for 5 min. The supernatant was removed and D-mannitol (MMG, 0.5 M (Sangon, Shanghai, China), 15 mM MgCl_2_ (Sangonz and 4 mM MES (Sangon (pH 7.5)) were added and incubated at room temperature for 8–10 min. Each experiment required 100 µL of protoplast solution.

A volume of 10 µL of plasmid that contained the target gene was mixed with 100 µL of protoplasts and 110 µL of a 40% solution of PEG 4000 (Yeasen, Shanghai, China). The mixture was then incubated at 22.5 °C for 15–20 min. The protoplasts were diluted with 1 mL of W5 solution and mixed well and then terminated. It was centrifuged at 400 rpm for 5 min and then the supernatant was discarded. Then, 1 mL of W5 was added to wash the pellet and then it was centrifuged twice. Another 1 mL of W5 was added, slowly mixed and then incubated overnight at 23 °C under low light. The solution was centrifuged at 400 rpm for 5 min. Supernatant was removed and the remaining pellet was imaged using laser confocal microscopy.

### 4.6. Disinfection and Cultivation of Arabidopsis

Sterilized seeds were placed separately in 1.5 mL centrifuge tubes (Sangon, Shanghai, China), washed with 75% ethanol (Sangon, Shanghai, China) for 1 min, washed with sterile distilled water for 3 min and then with 6% sodium hypochlorite for 8 min. The supernatant was discarded and the sample was washed with sterile distilled water for 3 min. The seeds were evenly spread on one-half MS solid culture medium, vernalized for 2 d at 4 °C and then incubated at 22 °C in 12 h/12 h (light/dark) for approximately 1 week. Normally developed seedlings were transplanted into a pot that contained a 1:1 ratio of vermiculite:nutrient soil (*v*/*v*). They were cultivated in a light culture room and then transformed with Agrobacterium tumefaciens at the peak flowering stage.

### 4.7. Construction of a Salt Stress Yeast Library for Suaeda salsa in Saline Soil

#### 4.7.1. Purification of mRNA

RNase-free H_2_O (Thermo Fisher) was added to an RNase-free PCR tube (Sangon) and the total RNA was diluted to 50 μL. Then, 50 µL of magnetic beads was added to the total sample of RNA and mixed. The sample was kept at 65 °C for 5 min and 25 °C for 5 min, so that the mRNA bound to the magnetic beads. The sample was placed on a magnetic rack and the supernatant was removed after the solution had clarified. The sample was removed from the magnetic holder, 200 μL of buffer was mixed in the L well and it was placed on a magnetic rack to remove the supernatant. The sample was removed from the magnetic holder and 50 μL of Tris-HCl buffer was added to resuspend the magnetic beads. It was kept at 80 °C for 2 min and the mRNA was eluted. Then, 50 µL binding buffer was added to the mRNA, mixed and incubated at room temperature for 5 min to enable the mRNA to bind to the magnetic beads. The sample was then placed on a magnetic rack and the supernatant was removed after the solution had clarified. Then, 10 µL RNase-free H_2_O (Thermo Fisher) was added and mixed well at 80 °C for 2 min. After the solution had cleared, the supernatant was carefully pipetted into a new RNase-free PCR tube. The RNA extraction detection map in Figure 16.

#### 4.7.2. Amplification and Purification of the cDNA

The reagent was added to the PCR product. A volume of 25 µL of 2 × PCR Mix, 2 µL of the three box primers P1/P2/P3-F, 2 μL of P4-R and 19 µL of ddH_2_O were mixed. The PCR reaction system was predenatured for 3 min at 95 °C, 95 °C for 15 s, annealed at 60 °C for 15 s, 72 °C for 6 min and incubated for 10 min at 72 °C for 30 cycles.

The PCR products were purified using DNA Clean Beads. The magnetic bead solution was stored at 4 °C for 30 min and then incubated at room temperature. The mixture was inverted to mix the magnetic bead solution. The magnetic bead solution that was 1.2-fold the sample volume was added to the DNA sample and thoroughly mixed by gently tapping the pipette 10 times. The solution was incubated at room temperature for 10 min to bind the DNA to the magnetic beads and then placed on a magnetic rack. The supernatant was removed after the solution had become clear and the sample was rinsed with 200 µL of 80% ethanol and incubated at room temperature for 30 s. The supernatant was removed and then inverted.

It was necessary to always keep the sample on a magnetic rack and the magnetic beads were opened and dried for 5–10 min at room temperature. The sample was removed from the magnetic holder and an appropriate amount of 30–50 μL of ddH_2_O that was free of nuclease was added.

The sample was incubated stationary at room temperature for 2 min and then placed on a magnetic rack to wait for the solution to clarify. The supernatant was aspirated into a new RNase-free PCR tube.

#### 4.7.3. Homogenization of cDNA and the Remove of Small Fragments

The reagents that were added to the PCR tube included 8 µL of 2× hybridization buffer (Thermo Fisher) and 1 mg of double-stranded cDNA, which was supplemented with ddH_2_O to a total volume of 16 μL. The solution was divided into four parts, kept in 2 mL PCR tubes at 98 °C for 2 min and incubated at 68 °C for 5 h.

Digestion with duplex-specific nuclease (DSN) was then performed. The following concentrations of DSN enzymes were prepared before the hybridization was completed. A volume of 1 mL DSN storage buffer was mixed with 1 mL of DSN enzyme and labeled as one-half DSN. A volume of 3 mL DSN storage buffer was mixed with 1 mL DSN enzyme and labeled as one-quarter DSN. The control was prepared by taking 1 mL of DSN storage buffer and marking it as the control. The DSN master buffer was preheated at 68 °C. Then, 4 µL of cDNA was added to 5 µL of 2 × DSN master buffer and 1 µL of DSN solution and mixed well. The mixture was incubated at 68 °C for 25 min and 10 mL of one of the four parts of the solution described above was added separately to each tube. The solution was mixed well after the DSN stop buffer had been added and incubated at room temperature for 5 min.

PCR amplification was performed after the DSN digestion. The reagents were added in sequence to the four PCR tubes as follows: 1 µL of the digested products (DSN, one-half DSN, one-quarter DSN and the control); 1 µL of 50× Advantage 2 Polymerase Mix; 1.5 µL of primer M1; 1 µL of 50× DNTP Mix; 10× Advantage 2 PCR Buffer 5 μL and 40.5 µL of ddH_2_O. The solutions were mixed well and labeled as DSNP1, one-half DSNP1, one-quarter DSNP1 and control lP1, respectively. All the components were gently mixed and briefly centrifuged before being placed in a preheated PCR machine.

The PCR reaction system was as follows: 95 °C for 1 min, 95 °C for 15 s, 60 °C for 20 s and 72 °C for 3 min. There were 11 cycles and 2 μL was taken after the reactions had been completed. Small fragments were removed according to the manufacturer’s instructions. The construction of cDNA membrane library of *S. salsa* in Figure 17.

#### 4.7.4. Homologous Recombination of cDNA and pPR3-N

The PCR tubes were placed on ice and the following solutions were added: 2 µL of ds cDNA, 4 µL of 5 × CE II Buffer, 2 µL of Exnase II and 50–200 ng of PPR3-N. Then, ddH_2_O was added to bring the volume to 20 μL. The tubes were incubated at 37 °C for 30 min and then placed on ice. Volumes of 20% of the products described above were added separately to sterile 1.5 mL centrifuge tubes and 2 µL of nucleic acid sedimentation aid was added along with 55 µL of 100% ethanol. The reaction volumes were then mixed well, incubated at −20 °C for 30 min and centrifuged at 13,000 rpm for 15 min. The supernatant was discarded and the sample was washed with 1 mL of 75% ethanol and then centrifuged at 8000 rpm for 2 min and this step was repeated. The pellet was dried at room temperature for 5–10 min and then dissolved and precipitated in 10 µL of TE buffer. The solution was aspirated and redissolved 60 times.

#### 4.7.5. Plasmid Transformation of *Escherichia coli*

A 2 mm electric shock cup was precooled on ice and 10 μL of recombinant product was added to 50 μL of competents. The solution was mixed and added to the electric shock cup. The shock meter was set to U = 2.5 kV. Immediately after the electric rotation, 2 mL of liquid SOC culture medium was added and the solution was transferred to a new 15 mL centrifuge tube, shaken and cultivated at 37 °C for 1 h. A volume of 10 µL of diluted bacterial solution was diluted 10,000-fold and a 100 μL aliquot was coated onto LB plates that contained 50 mg/L Amp and kept upside down at 37 °C overnight.

#### 4.7.6. Library Identification

The library capacity (CFU·mL^−1^) = titer × volume of the bacterial solution. The total number of clones (CFU) = storage capacity × total bacterial volume in the experiment.

The clones were identified by randomly selecting 24 clones as templates and then adding them to the following components: 1 µL pPR3-N-F, 1 µL PPR3-N-R, 10 µL 2 × PCR Mix (Thermo Fisher) and 8 µL ddH_2_O (Thermo Fisher). The PCR reaction system was as follows: 95 °C for 3 min, 95 °C for 15 s, 55 °C for 15 s, 72 °C for 1 min and 72 °C for 5 min. A total of 25 cycles were completed and 5 µL was removed after the reaction was complete for detection with electrophoresis.

#### 4.7.7. Library Plasmid Extraction

First, 100 µL of library bacterial solution was inoculated into a triangular flask that contained 100 mL lysogeny broth and cultured to an OD_600_ of 1.0 at 30 °C and 220 rpm. The bacterial cells were collected using a 50 mL centrifuge tube and centrifuged at 8000 rpm for 3 min at room temperature to remove supernatant. Then, 10 mL of solution P1 containing RNase A was added to suspend the precipitate which was then thoroughly vortexed. Then, 10 mL of solution P2 was added and the solution was pipetted up and down 6–8 times at room temperature for 5 min. Next, 10 mL of solution P4 was added and treated as described for solution P2 before incubation at room temperature for 10 min. The solution was centrifuged at 8000 rpm for 10 min and filtered into a new centrifuge tube. Then, 0.3 times the volume of isopropanol was added, mixed and transferred to an adsorption column at a rate of 10 mL/time. The mixture was centrifuged at 8000 rpm for 2 min and waste liquid was poured off until all the solutions had passed through the column. Then, 10 mL of deproteinized solution was added to the adsorption column CP6 and centrifuged at 8000 rpm for 2 min, and the waste liquid was finally discarded. Then, 10 mL of rinsing solution was added to the adsorption column CP6 and centrifuged at 8000 rpm at room temperature for 2 min, and the waste liquid was finally discarded. Then, 10 mL of rinsing solution was added to adsorption column CP6 and the steps were repeated. The sample was centrifuged at 8000 rpm for 5 min at room temperature and then incubated at room temperature for several minutes until the ethanol had been thoroughly air-dried. Then, 1–2 mL of eluent was added to the center of the membrane, incubated at room temperature for 5 min and centrifuged at 8000 rpm for 5 min. The solution was transferred to a new 1.5 mL centrifuge tube and stored at −20 °C to obtain the library plasmid.

### 4.8. Yeast Transformation and Detection of Self-Activation

#### 4.8.1. Transfer of pBT3-N-SsNRT1.1C and pBT3-N into NMY51

A single colony of NMY51 from the YPDA plate was inoculated onto 4 mL of YPDA liquid culture medium at 30 °C, 225 rpm and shaken for 18–20 h (overnight) until the OD_600_ > 1.5. The colony was transferred to 50 mL of liquid YPDA and incubated at an initial OD_600_ of 0.2, 30 °C, 225 rpm and shaken for 4–5 h until the OD_600_ was 0.6. The culture was centrifuged at 4000 rpm for 5 min and the bacteria were resuspended with 20 mL of sterile water, mixed well and centrifuged again as described above. The supernatant was discarded and the bacteria were suspended in 5 mL of 0.1 M lithium acetate (LiAc), mixed well and centrifuged at 4000 rpm for 5 min. The supernatant was discarded and the bacteria were resuspended in 500 µL of 0.1 M LiAc, mixed well and divided into 1.5 mL centrifuge tubes, which each contained 50 µL. Then, 240 µL of 50% PEG3350, 36 µL of 1 M LiAc, 5 µL of 20 mg/mL single-stranded DNA (ssDNA) and 5 µL of plasmid DNA were added to each 1.5 mL centrifuge tube and shaken vigorously for approximately 1 min until completely mixed. The solution was incubated at 30 °C for 30 min and then subjected to heat shock for 25 min at 42 °C, then cooled at 30 °C for 30 min. The solution was then centrifuged for 5 min at 4000 rpm and the supernatant was discarded. A volume of 200 µL of sterile water was used to suspend each transformant, which was mixed as gently as possible and applied to the corresponding defect type screening plate. The cultures were incubated at 30 °C for 4 d.

#### 4.8.2. Detection of Self-Activation

pBT3-N-SsNRT1.1C and pBT3-N were transformed into NMY51, which was then inoculated onto SD/Leu plates and incubated at 30 °C for 3–5 d. Six spots were randomly picked and primers were used for PCR verification. After three correct clones had been mixed, the OD_600_ was adjusted to 0.002. The clones were diluted into a gradient of 10^0^, 10^−1^, 10^−2^ and dotted onto the corresponding defective plate that contained different concentrations of 0, 10, 20, 30, 40, 50, 75 and 100 mM of 3-amino-1,2,4-triazole (3-AT) and incubated at 30 °C for 3 d. 3-AT competitively inhibits yeast HIS3 protein synthesis, which is used to inhibit the leakage expression of *HIS3*.

#### 4.8.3. Yeast Library Screening

The receptive state was prepared using the NMY51 yeast strain that contained pBT3-N-SsNRT1.1C bait plasmid, which was transferred into the membrane yeast cDNA library prepared from *S. salsa* that had been treated with NaCl and coated on the SD-TLH and SD-TLHA screening plates of 0 mM 3-AT.

#### 4.8.4. Library DNA Transformation

A single bacterial strain from the SD-L plate was transferred to 50 mL of liquid SD-L and shaken for 24 h at 225 rpm and 30 °C. A volume of 500 mL of YPDA was transferred with an initial OD_600_ of 0.2 and shaken at 225 rpm for 4–5 h at 30 °C until the OD_600_ was 0.6. The cells were collected by centrifugation for 5 min at 4000 rpm and, after adding 30 mL of sterile water to resuspend bacteria, mixed well and centrifuged again as described above and the supernatant was discarded. The bacteria were resuspended in 20 mL of 0.1 M LiAc, mixed well and centrifuged as described above and then the supernatant was discarded. The bacteria were resuspended in 10 mL of 0.1 M LiAc, mixed well and centrifuged as described above and then the supernatant was discarded. A volume of 9.6 mL of 50% PEG3350, 1.44 mL of 1 M LiAc, 300 µL of ssDNA (10 mg/mL), 25 ug of library plasmid DNA were added to the centrifuge tube and shaken vigorously for approximately 1 min until completely mixed. The solution was incubated at 30 °C for 30 min and then subjected to heat shock at 42 °C for 25 min. The cells were then incubated at 30 °C for 1 hr to recover and then centrifuged as described above. The supernatant was discarded, and the pellet of cells was resuspended with 6 mL of sterile water and mixed as gently as possible. An aliquot of 20 µL was diluted and applied to an SD-TL plate to test the library conversion efficiency. The rest of the cells were coated on 10 SD-TLH and 10 SD-TLHA flat plates each. The plates were incubated at 30 °C for 3–7 days and the colony growth was observed.

#### 4.8.5. Positive Clone Identification and Sequencing Comparison

To understand the specific information of the monoclones, we amplified them, sequenced the DNA and compared and analyzed the sequences using BLAST.

#### 4.8.6. Rotational Validation of the Positive Yeast Clones

The positive clone transformants grown on the SD-TLH-deficient plates were diluted with sterile water at concentrations of 10^0^, 10^−1^, 10^−2^, then transferred to the SD-TL-, SD-TLH- and SD-TLHA-deficient plates, respectively. The colonies were incubated at 30 °C for 3–4 days.

### 4.9. Double Molecule Fluorescence Complementary Experiment (BiFC) to Detect the Protein Interactions

The protein interaction between SsNRT1.1C and SsHINT1 is utilized as an example.

The CDS sequence of *SsHINT1* was ligated to the C-terminus of YFP using a one-step recombination method, while *SsNRT1.1C* was ligated to the N-terminus of YFP. Finally, SsNRT1.1C-YFPN and SsHINT1-YFPC were obtained. The experimental method and steps are the same as those of the subcellular localization in Section 2.5.

## Figures and Tables

**Figure 1 ijms-24-12761-f001:**
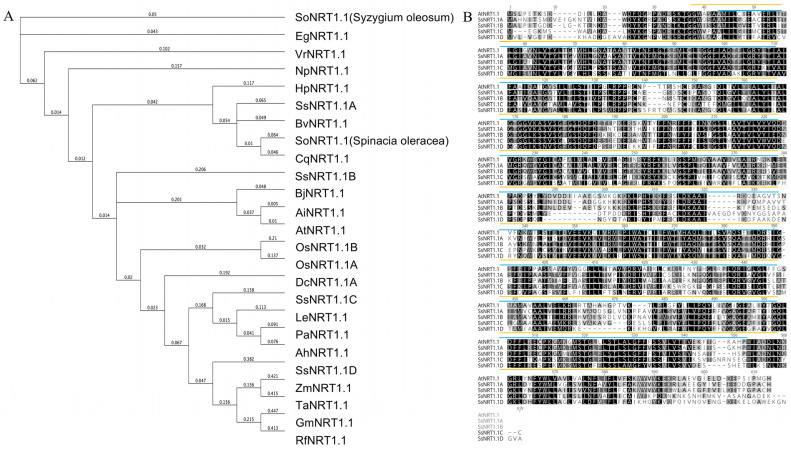
Analysis of the protein sequence of SsNRT1.1. (**A**) Phylogenetic analysis of SsNRT1.1 with orthologous NRT1.1 proteins from other plant species. Except for SsNRT1.1A, SsNRT1.1B, SsNRT1.1C and SsNRT1.1D of *Suaeda salsa*, SoNRT1.1 of *Syzygium oleosum*, EgNRT1.1 of *Eucalyptus grandis*, VrNRT1.1 of *Vitis riparia*, NpNRT1.1 of *Nicotiana plumbaginifolia*, HpNRT1.1 of *Heliosperma pusillum*, BvNRT1.1 of *Beta vulgaris*, SoNRT1.1 of *Spinacia oleracea*, CqNRT1.1 of *Chenopodium quinoa*, BjNRT1.1 of *Brassica juncea*, AlNRT1.1 of *Arabidopsis lyrate*, AtNRT1.1 of *Arabidopsis thaliana*, OsNRT1.1A and OsNRT1.1B of *Oryza sativaL*, DcNRT1.1A of *Dendrobium catenatum*, LeNRT1.1 of *Lithospermum erythrorhizon*, PaNRT1.1 of *Prosopis alba*, AhNRT1.1 of *Arachis hypogaea*, ZmNRT1.1 of *Zea mays*, TaNRT1.1 of Triticum aestivum, GmNRT1.1 of *Glycine max*, RfNRT1.1 of *Rhododendron fortune* are shown. (**B**) Amino acid sequence alignment and analysis of SsNRT1.1s and AtNRT1.1. Gray boxes exhibit where physically and chemically similar amino acids have been replaced within different NRT proteins. Different colors are used to underline the distinct predicted structures. NRT, nitrate transporter.

**Figure 2 ijms-24-12761-f002:**
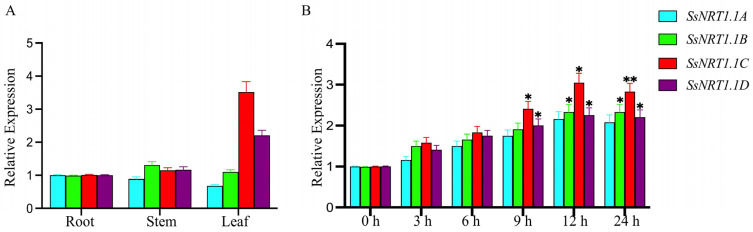
Expression of the *SsNRT1.1* genes in *Suaeda salsa*. (**A**) Four *SsNRT1.1* genes’ expression in different tissues and the *SsNRT1.1C* gene expression is highest in the leaf; (**B**) four genes’ expression in six-week-old leaves of *S. salsa* following treatment with 400 mM NaCl and *SsNRT1.1C* gene expression is highest. Data are presented as means (±SD) of three biological replicates. Asterisks above the data bars indicates a significant difference (two-tailed *t*-test * *p* < 0.05 ** *p* < 0.01) or no significant difference, respectively.

**Figure 3 ijms-24-12761-f003:**
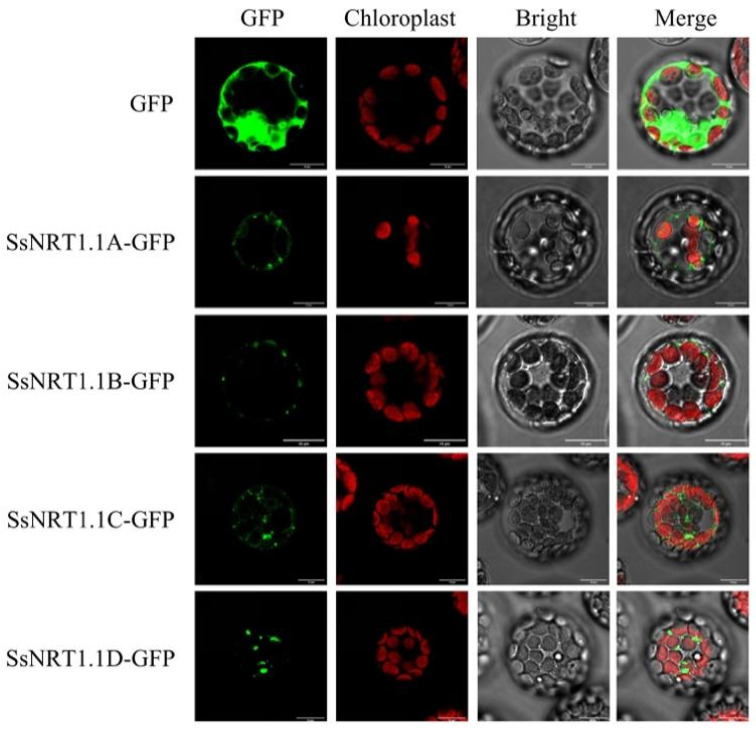
Subcellular localization of the SsNRT1.1s-GFP fusion proteins. We constructed 4 SsNRT1.1 protein GFP vectors and observed subcellular localization using *Arabidopsis* protoplasts. Four SsNRT1.1s-GFP are all located in the endoplasmic reticulum (ER). GFP is positive; green fluorescence excitation wavelength: 488 nm; chloroplast autofluorescence excitation wavelength: 640 nm. Bar, 10 μm. GFP, green fluorescent protein.

**Figure 4 ijms-24-12761-f004:**
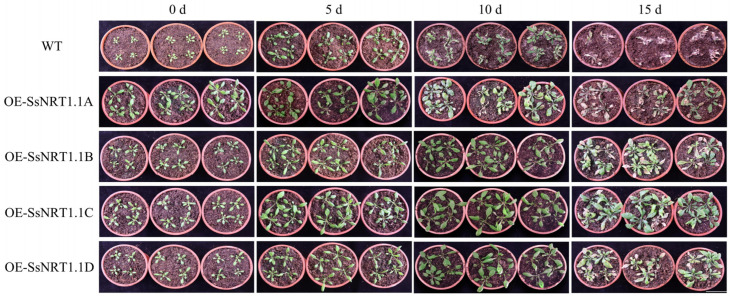
The phenotype of transgenic *Arabidopsis* under salt stress. Thirty-day-old *Arabidopsis* plants under 250 mM NaCl salt stress were chosen. Days 0, 5, 10 and 15 were selected for statistics. Before salt treatment, the transgenic plants were bigger than WT. During salt treatment, the transgenic leaf color was more greenish and survival rates were higher. At 15 d, the WT were no longer growing and most transgenic plants continued to grow. Bar, 40 mm.

**Figure 5 ijms-24-12761-f005:**
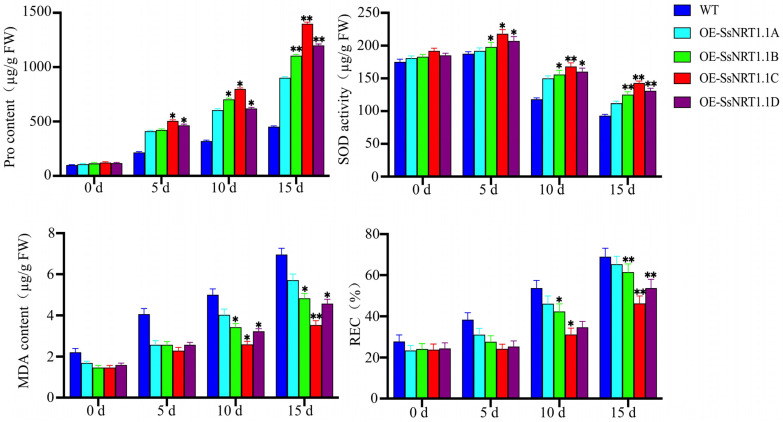
Physiological measurement of *SsNRT1.1s* transgenic *Arabidopsis* under salt stress. Pro, SOD, MDA and REC were chosen for the physiological detection index in *Arabidopsis* salt stress. Pro contents were increased during salt treatment and OE-SsNRT1.1C had the highest. MDA and REC were increased during salt treatment and WT had the highest, OE-SsNRT1.1C had the lowest compared to other transgenic *Arabidopsis*. SOD activity increased at 5 d and then decreased, OE-SsNRT1.1C’s SOD activity variation was minimal. Data are presented as means (±SD) of three biological replicates. Asterisks above the data bars indicates a significant difference (two-tailed *t*-test * *p* < 0.05 ** *p* < 0.01) or no significant difference, respectively.

**Figure 6 ijms-24-12761-f006:**
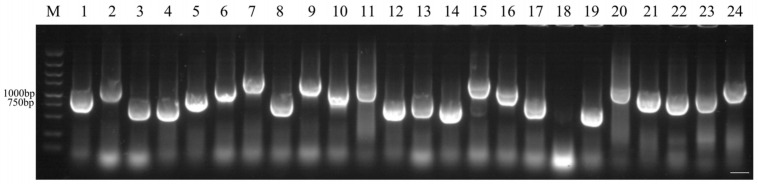
PCR analysis of the cDNA fragments in *S. salsa* yeast two-hybrid library. Twenty-four monoclones were randomly selected for PCR detection and agarose gel electrophoresis, and finally twenty-three bands were obtained. Bar, 3 mm.

**Figure 7 ijms-24-12761-f007:**
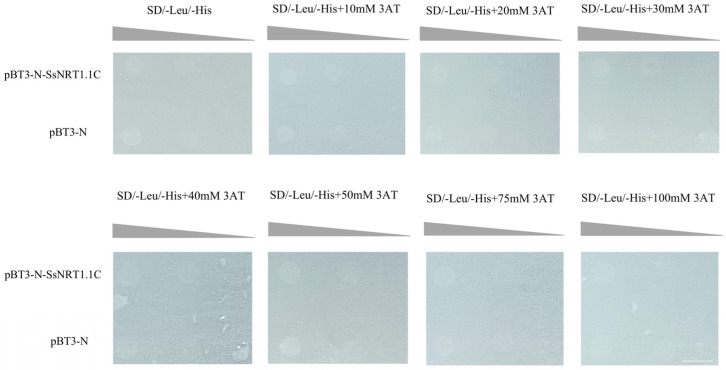
SsNRT1.1C bait carrier self-activation detection. The bait expression vector pBT3-N-SsNRT1.1C was constructed and placed in an SD/-Leu/-His Petri dish. Then, 0, 10, 20, 30, 40, 50, 75 and 100 mM 3AT were added to detect the expression of *HIS3* gene. Bar, 4 mm.

**Figure 8 ijms-24-12761-f008:**
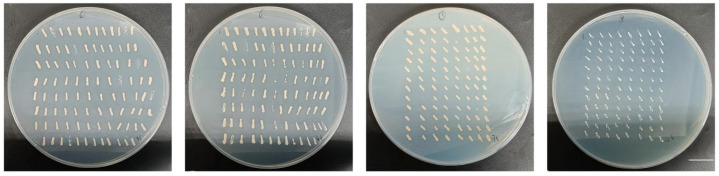
Screen of pBT3-N-SsNRT1.1C on SD-TLH plate of *S. salsa* yeast two-hybrid library. The *S. salsa* cDNA membrane library plasmid was transferred to NMY51 yeast strain, and 384 monoclones were finally screened on SD-TLH selection plates at 0 mM 3AT. Bar, 18 mm.

**Figure 9 ijms-24-12761-f009:**
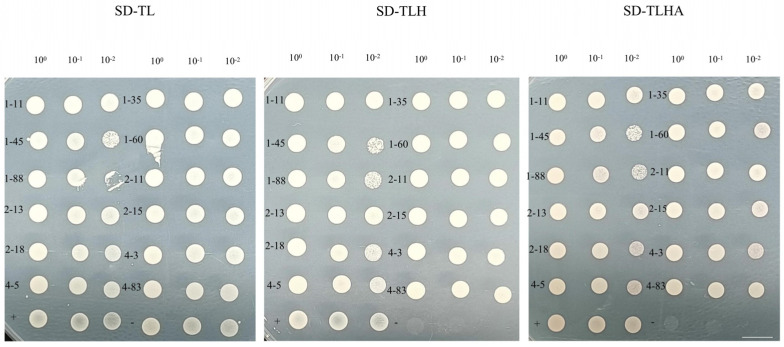
Rotational validation of yeast positive clones on SD-TL-, SD-TLH- and SD-TLHA-defective plates. 1–11, 1–35, 1–45, 1–60, 1–88, 2–11, 2–13, 2–15, 2–18, 4–3, 4–5, 4–83 correspond to the monoclonal numbers screened by the previous experiment. Bar, 4 mm.

**Figure 10 ijms-24-12761-f010:**
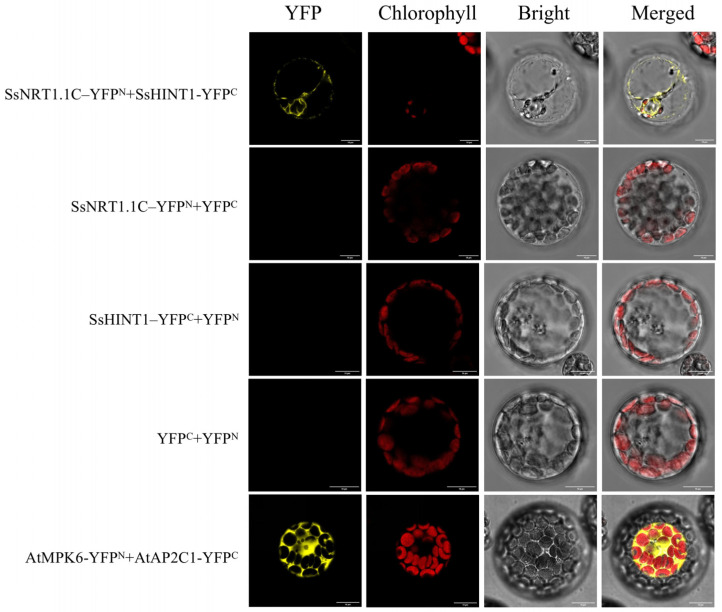
SsNRT1.1C interacts with SsHINT1 in BiFC experiments with *Arabidopsis* protoplasts. SsNRT1.1C and SsHINT1 YFP vectors were constructed, SsNRT1.1C-YFP^N^ + SsHINT1-YFP^C^ results show they interacted. SsNRT1.1C-YFP^N^ + YFP^C^, SsHINT1-YFP^C^ + YFP^N^, YFP^C^ + YFP^N^ are negative controls and did not interact. AtMPK6-YFP^N^ + AtAP2C1-YFP^C^ is positive control and interacted. Bar, 10 μm.

**Figure 11 ijms-24-12761-f011:**
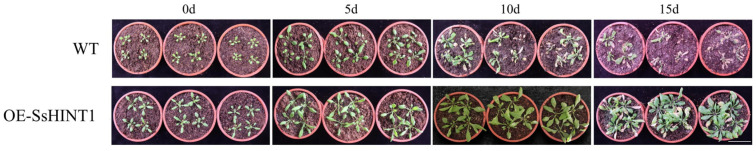
Salt stress treatment of *Arabidopsis* plants. Thirty-day-old *Arabidopsis* under 250 mM NaCl salt treatment were chosen. Days 0, 5, 10 and 15 were selected for statistics. Before salt treatment, the 0 d transgenic plants were bigger than WT. During salt treatment, the transgenic leaf color was more greenish and survival rates were higher. Bar, 40 mm.

**Figure 12 ijms-24-12761-f012:**
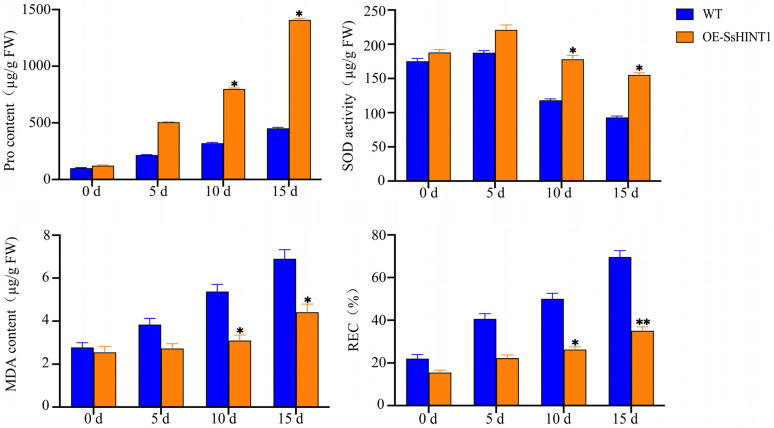
*SsHINT1* transgenic *Arabidopsis* physiological detection under salt stress. Pro, SOD, MDA and REC were chosen as a physiological detection index in *Arabidopsis* salt stress. Pro content was increased during salt treatment and OE-SsHINT1 had the highest. MDA and REC were increased during salt treatment and WT had the highest. SOD activity increased at 5 d after salt treatment and then decreased and the WT had a greater decline. Data are presented as means (±SD) of three biological replicates. Asterisks above the data bars indicates a significant difference (two-tailed *t*-test * *p* < 0.05 ** *p* < 0.01) or no significant difference, respectively.

**Figure 13 ijms-24-12761-f013:**
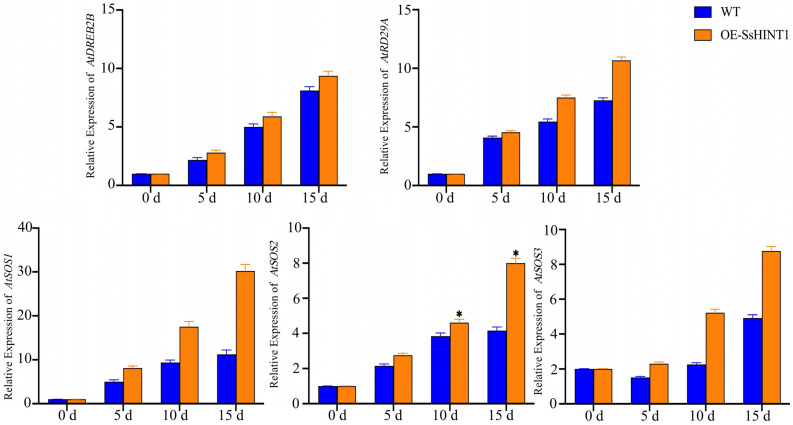
Expression of related genes under salt stress in *Arabidopsis. AtDREB2B*, *AtRD29A*, *AtSOS1*, *AtSOS2*, AtSOS3 were selected for qRT-PCR, which are salt tolerance genes. The expression of these genes began to rise with time and OE-SsHINT1’s expression was higher than in the WT. In particular, *AtSOS1/2/3* expression was higher than in the WT. Data are presented as means (±SD) of three biological replicates. Asterisks above the data bars indicates a significant difference (two-tailed *t*-test * *p* < 0.05) or no significant difference, respectively.

**Figure 14 ijms-24-12761-f014:**
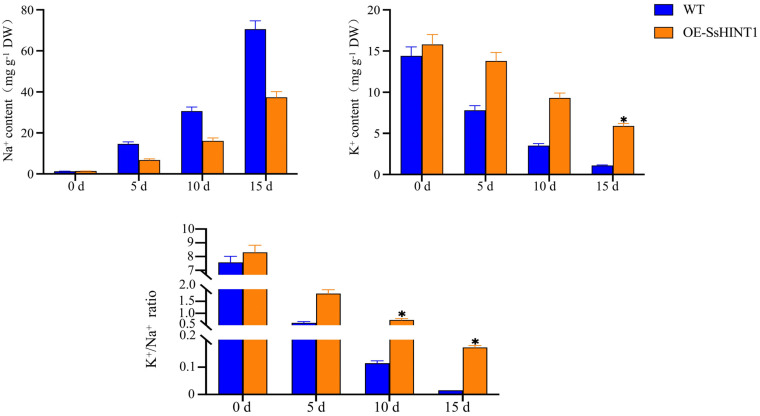
Na^+^ and K^+^ contents in *SsHINT1* transgenic *Arabidopsis.* Days 0, 5, 10, 15 were chosen to detect the Na^+^ and K^+^ content. The Na^+^ content was increased and K^+^ content was decreased. The K^+^/Na^+^ ratio was decreased, but OE-SsHINT1’s ratio was higher than in WT, which became more evident over time. Data are presented as means (±SD) of three biological replicates. Asterisks above the data bars indicates a significant difference (two-tailed *t*-test * *p* < 0.05) or no significant difference, respectively.

**Figure 15 ijms-24-12761-f015:**
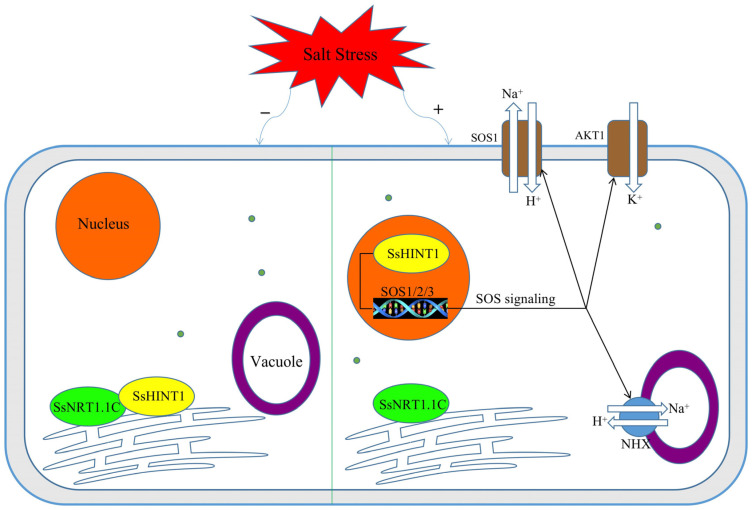
Functional prediction of *NRT1.1C* and *HINT1* genes under salt stress. The left shows that, under normal conditions, the NRT1.1C and HINT1 proteins are in the ER and interact. When the plant suffers salt stress, the cells will take on the state on the right. The SsHINT1 protein leaves the ER, enters the nucleus and activates SOS1/2/3. After the SOS signal is activated, the expression of related genes is increased, the Na^+^ content is reduced and the K^+^ content is increased.

**Figure 16 ijms-24-12761-f016:**
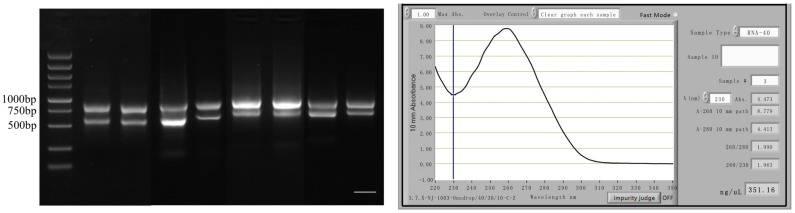
RNA extraction. The highest expression period was selected for RNA extraction, the gel electrophoresis after RNA extraction is shown on the left and the RNA concentration detected by NanoDrop is shown on the right. Bar, 4 mm.

**Figure 17 ijms-24-12761-f017:**
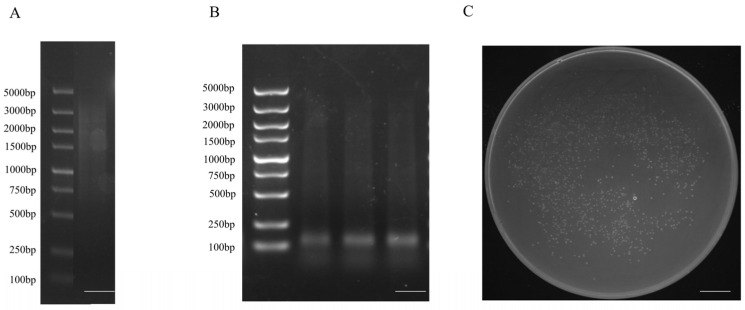
Construction of cDNA membrane library of *S. salsa*: (**A**) RNA extraction, M: Marker 2 kb plus, 1–4: *S. salsa* sample RNA; (**B**) ds cDNA synthesis, M: Marker 2 kb plus; 1: P1-F/P4-R amplification of ds cDNA; 2: P2-F/P4-R amplification of ds cDNA; 3: P3-F/P4-R amplification of ds cDNA; (**C**) ds cDNA purification; M: Marker 2 kb plus; 1: Mixed homogenization and removal of small fragments of three ds cDNAs; (**C**) cDNA library capacity of *S. salsa* yeast. Bar, 2 mm; Bar, 3 mm; Bar, 12 mm.

**Table 1 ijms-24-12761-t001:** Analysis of the physical and chemical properties of the *SsNRT1.1* genes.

Gene Name	CDS Length/bp	Amino Acid Number	Molecular Weight/kDa	pI	Instability Coefficient	Hydrophilicity
*SsNRT1.1A*	1791	596	65.777	8.71	36.93	0.391
*SsNRT1.1B*	1782	593	65.186	7.54	33.37	0.398
*SsNRT1.1C*	1755	584	64.006	8.87	26.57	0.363
*SsNRT1.1D*	1746	581	64.185	8.51	25.00	0.291

**Table 2 ijms-24-12761-t002:** Primers used in the experiment (F: Forward primers; R: Reverse primers).

Primer Name	Sequence
OE-SsNRT1.1A-F	ACTAGGGTCTCGCACCATGGCTCTTCCTATAACAGGCGAT
OE-SsNRT1.1A-R	ACTAGGGTCTCTACCGTCAATGGCAGGCTGGGGT
OE-SsNRT1.1B-F	ACTAGGGTCTCGCACCATGGCTCATAATATTACTAGTATGG
OE-SsNRT1.1B-R	ACTAGGGTCTCTACCGTTAGTGACAAGCCGGACCATCGTC
OE-SsNRT1.1C-F	ACTAGGGTCTCGCACCATGGAGGGGAAGATGAGTTGGG
OE-SsNRT1.1C-R	ACTAGGGTCTCTACCGCTAGCATTTTTCATCAGCACCATTA
OE-SsNRT1.1D-F	ACTAGGGTCTCGCACCATGGTTTTAGTTGGAGAGTTTGA
OE-SsNRT1.1D-R	ACTAGGGTCTCTACCGTTAGGCAACTCCATTGCCTTTTTCCC
OE-SsHINT1-F	ACTAGGGTCTCGCACCATGAGCAATGACAATCTTATAA
OE-SsHINT1-R	ACTAGGGTCTCTACCCAAATGAACTGGCCCCCTGGCT
CDS III/3′ PCR Primer	AAGCAGTGGTATCAACGCAGAGTGGCCATTATGGCCGGG
SMART IV Oligonucleotide	ATTCTAGAGGCCGAGGCGGCCGACATGTTTTTTTTTTTTTTTTTTTTTTTTTTTTTTVN
P1-F	TACGATGTTCCAGATTACGCTGGATCCAAGCAGTGGTATCAACGCAGAGTGG
P2-R	TACGATGTTCCAGATTACGCTGGATCCAAAGCAGTGGTATCAACGCAGAGTGG
P3-F	TACGATGTTCCAGATTACGCTGGATCCAAAAGCAGTGGTATCAACGCAGAGTGG
P4-R	GGTATCGATAAGCTTGATATCGAATTCCTAGAGGCCGAGGCGGCCGACATG
Prime M1	AAGCAGTGGTATCAACGCAGAGT
pPR3-N-F	CGGTAAAACCGGAACATTGGA
pPR3-N-R	ACTTCAGGTTGTCTAACTCCT
GFP-SsNRT1.1A-F	ACTAGGGTCTCGCTCCATGGCTCTTCCTATAACAGGCGAT
GFP-SsNRT1.1A-R	ACTAGGGTCTCTACCGTCAATGGCAGGCTGGGGT
GFP-SsNRT1.1B-F	ACTAGGGTCTCGCTCCATGGCTCATAATATTACTAGTATGG
GFP-SsNRT1.1B-R	ACTAGGGTCTCTTTAGTGACAAGCCGGACCATCGTC
GFP-SsNRT1.1C-F	ACTAGGGTCTCGCTCCATGGAGGGGAAGATGAGTTGGG
GFP-SsNRT1.1C-R	ACTAGGGTCTCTACCGCTAGCATTTTTCATCAGCACCATTA
GFP-SsNRT1.1D-F	ACTAGGGTCTCGCTCCATGGTTTTAGTTGGAGAGTTTGATAAACATGCT
GFP-SsNRT1.1D-R	ACTAGGGTCTCTACCGTTAGGCAACTCCATTGCCTTTTTCCC
ACTIN-F	TGGTGTCATGGTTGGGATG
ACTIN-R	CACCACTGAGCACAATGTTAC
qRT-SsNRT1.1A-F	AAGCTTGGGGTCACTAGCAG
qRT-SsNRT1.1A-R	CTGCTGTTTTTGTCGCTGCT
qRT-SsNRT1.1B-F	AGGTTTTCCGGCTAATCGCA
qRT-SsNRT1.1B-R	AGGTTTTCCGGCTAATCGCA
qRT-SsNRT1.1C-F	GCTTCGACCACCACCATGTA
qRT-SsNRT1.1C-R	GCTTCGACCACCACCATGTA
qRT-SsNRT1.1D-F	TACTGGTGGATGGCTTGCTG
qRT-SsNRT1.1D-R	TACTGGTGGATGGCTTGCTG
qRT-SsHINT1-F	ACAGGAAGGCCTTGACGATG
qRT-SsHINT1-R	TTTGGCGTCCTCCAAGAAGG
pBT3-SsNRT1.1C-F	AATTCCTGCAGGGCCATTACGGCCATGGAGGGGAAGATGAGTTG
pBT3-SsNRT1.1C-R	CTACTTACCATGGGGCCGAGGCGGCCATGGTGCTGATGAAAAATGC
BiFC-SsNRT1.1C-F	ACTAGGGTCTCGCACCATGGAGGGGAAGATGAGTTGGGC
BiFC-SsNRT1.1C-R	ACTAGGGTCTCTCGGAGCATTTTTCATCAGCACCATTAGCACTAGC
BiFC-SsHINT1-F	ACTAGGGTCTCGCACCATGGCTCAATCAGCGAAACCAATCAC
BiFC-SsHINT1- R	ACTAGGGTCTCTCGGAGACATAAACACCGGTTGATAGGAGTAAAAATAAGGAT
At SOS1- F	CCTCGAGAAGGTTGGCTTGT
At SOS1-R	ATGCAGGAGGAAGAGCAACC
At SOS2-F	GCTCATTGTCACTGCAAGGG
At SOS2-R	ACTCCTTCCTGAGGCAATGC
At SOS3-F	GTCACGCCATTCACGGTAGA
At SOS3-R	GTCACGCCATTCACGGTAGA
AtRD22-F	TTTGGAATACGCGGGACACA
AtRD22-R	AAGGAACCATCGTGCAGCTT
AtRD29A-F	TGAAGCCAGAATCGCCACAT
AtRD29A-R	TGAAGCCAGAATCGCCACAT

**Table 3 ijms-24-12761-t003:** qRT-PCR reaction system (F: Forward primers; R: Reverse primers).

Component	Volume (µL)
cDNA	3–5
2 × qPCR SuperMix	10
Primer-F (10 μM)	1
Primer-R (10 μM)	1
ddH_2_O	3–5

Note: The qRT-PCR amplification reaction program included predenaturation at 94 °C for 3 min, denaturation at 94 °C for 15 s, annealing at 58 °C for 15 s and extension at 72 °C for 10 s. There were 40 cycles of denaturation, annealing and extension. The 2^−ΔΔT^ method was used to analyze the experimental qRT-PCR data.

## Data Availability

The original contributions presented in this study are publicly available. The data can be found in the National Center for Biotechnology Information (NCBI) BioProject database under accession numbers PRJNA527358 and PRJNA512222.

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
