# Peer review of "Suaeda salsa NRT1.1 Is Involved in the Regulation of Tolerance to Salt Stress in Transgenic Arabidopsis"

_ijms, 2023, doi:10.3390/ijms241612761_

Round 1
Reviewer 1 Report
Dear Authors,
Xiong et al. is a beautiful piece of work. The authors have substantially validated each and every experiment. The write up is clean and self-explanatory. The model at the end is so self-explanatory and interesting. The findings have been validated and shown at each step be in a cDNA prep or Y2H assay, that I cannot ask for more experiments.
Just a suggestion, the first two paragraphs in the discussion section is a little bit of loosely bound, if that can be sorted.
Reviewer 2 Report
Article Suaeda salsa NRT1.1 is Involved in the Regulation of Tolerance
to Salt Stress in Transgenic Arabidopsis by Yi Xionga,Saisai Wanga,Cuijie Cuia,Xiaoyan Wua,Jianbo Zhua describes the interaction of a heterologous product with native instruments for regulation of salt response sensitivity through Na/K regulation.
The authors believe that this system can be stable and actually work for different species when localized in the EPR.
The manuscript is not designed according to the rules (literary sources are not numbered and references in the form of surnames), although it contains the necessary sections, except for statistics.
There are many questions related to the concept itself, which is represented by the resulting figure15. The drawing is sloppy and does not contain labels of the compartments that the authors probably wanted to present.
We are seeing some very embarrassing moments that are connected with logic.
1. I see a cell, and not two different cells, since they have one cell wall, which the authors probably tried to depict.
2. I don’t know what a minus is in understanding stress, that is, what will happen after it is canceled or before it occurs?
3. I do not see plasmolysis or other signs of stress - how can I judge it? 250mM NaCl salt stress?
For some reason, the figure captions do not contain complete information, for example: Figure 11. Salt stress treatment of Arabidopsis plants.
Or: Figure 10. SsNRT1.1C interact with SsHINT1 in BiFC experiments with Arabidopsis protoplasts Bars 10 μm.
It must be reflected in deciphered form: what it is, how it was obtained (method), what the colors mean, the rulers and / or magnification must be clear and visible.
Or: Figure 5. Physiological measurement of SsNRT1.1s transgenic Arabidopsis under salt stress.
See also: Figure 2. Expression of the SsNRT1.1 genes in Suaeda salsa. (A) expression in different tissues; (B)144
Expression in the leaves following treatment with 400 mM NaCl.
Where is the information about the value of p ? Why are the drawings small and not in color, poorly distinguishable columns and reliability.
on p. 88, the names of plants are highlighted in different inclinations of the text, what is the meaning?
Figures 7-9 are not informative and may be passed down to the application or need to be processed.
On fig 17 c - required bar
line 380-200mM and even 300mM are not stressful for this plant - this is an effect
Reagents and equipment require manufacturer identification even if RNase-free H2O ; 2x Hybridization 479
buffer; PCR machine...
Section 4.7.6.
Bacterial Solution... write clearly
section 4.5.
ºC - remove degree underline
Table 2. Primers used in the experiment. (decryption F and R are also needed.
Careful study of the entire text is required for its evaluation.
The question of localization remains mysterious, as does the hieroglyph in the center of figure 15 near the word SOS1
Round 2
Reviewer 2 Report
Article Suaeda salsa NRT1.1 is Involved in the Regulation of Tolerance
to Salt Stress in Transgenic Arabidopsis by Yi Xionga, Saisai Wanga,Cuijie Cuia,Xiaoyan Wua,Jianbo Zhu has been improved.
The article can be accepted, but I recommend changing the black and white histograms to color ones. This should improve the quality of the feed.